# Preparation of Azoxystrobin-Zinc Metal–Organic Framework/Biomass Charcoal Composite Materials and Application in the Prevention and Control of Gray Mold in Tomato

**DOI:** 10.3390/ijms242115609

**Published:** 2023-10-26

**Authors:** Xiao Han, Yinjie Qian, Jiapeng Li, Zhongkai Zhang, Jinbo Guo, Ning Zhang, Longyu Liu, Zhiqiang Cheng, Xiaobin Yu

**Affiliations:** 1College of Plant Protection, Jilin Agricultural University, Changchun 130118, China; hanxiao17614311202@163.com (X.H.); 20210219@mails.jlau.edu.cn (Y.Q.); zzk199707@163.com (Z.Z.); guojinbo524@163.com (J.G.); zn19961310@163.com (N.Z.); lly000512@163.com (L.L.); 2College of Resources and Environment, Jilin Agricultural University, Changchun 130118, China; lijiapeng52@163.com

**Keywords:** AZOX-ZIF-8/BC, *Botrytis cinerea*, fungicide, nano fungicide, pH-responsive release

## Abstract

In order to reduce the use of fungicide and ensure food safety, it is necessary to develop fungicide with low toxicity and high efficiency to reduce residues. Azoxystrobin (AZOX), which is derived from mushrooms, is an excellent choice. However, conventional AZOX release is difficult to regulate. In this paper, a pH-responsive fungicide delivery system for the preparation of AZOX by impregnation method was reported. The Zinc metal–organic framework/Biomass charcoal (ZIF-8/BC) support was first prepared, and subsequently, the AZOX-ZIF-8/BC nano fungicide was prepared by adsorption of AZOX onto ZIF-8/BC by dipping. Gray mold, caused by *Botrytis cinerea*, is one of the most important crop diseases worldwide. AZOX-ZIF-8/BC could respond to oxalic acid produced by *Botrytis cinerea* to release loaded AZOX. When pH = 4.8, it was 48.42% faster than when pH = 8.2. The loading of AZOX on ZIF-8/BC was 19.83%. In vitro and pot experiments showed that AZOX-ZIF-8/BC had significant fungicidal activity, and 300 mg/L concentration of AZOX-ZIF-8-BC could be considered as a safe and effective control of *Botrytis cinerea*. The above results indicated that the prepared AZOX-ZIF-8/BC not only exhibited good drug efficacy but also demonstrated pH-responsive fungicide release.

## 1. Introduction

The sustainable development of agriculture heavily relies on the widespread implementation of fungicide to mitigate crop yield losses and enhance productivity and product quality [1]. The prevalence of vegetable fungal diseases is on the rise, and gray mold of tomato caused by *Botrytis cinerea* is a serious threat to crop yield and quality [2,3]. *Botrytis cinerea* causes yield loss of more than 20% and economic loss of tens of billions of yuan each year [4]. It poses a significant threat to the productivity of tomato crops by infiltrating multiple plant components, including stems, leaves, and fruits, thereby compromising food safety [5]. *Botrytis cinerea* has a strong environmental adaptability and produces spores that can easily infect nearby plants [6].After harvest, this pathogen can persist as mycelium, with or without conidia and sclerotia, in crop wastes [7]. Unfortunately, to date, there are still no reports on resistant tomato varieties [8]. The most common and conservative way to control the devastating damage of *Botrytis cinerea* to agriculture is to use large quantities of chemical fungicides, such as thiabendazole, phenylpyrrole, benomyl, and flunidoxime [9]. However, the excessive utilization of fungicide can readily engender drug resistance in certain pathogenic microorganisms [10]. Hence, it is imperative to devise novel approaches for curtailing fungicide usage to avert this grave phenomenon.

The compound AZOX (Figure 1) was initially derived from mushrooms and has gained widespread recognition as a highly efficacious methoxyacrylate fungicide, offering robust crop protection against diverse pathogenic fungi on a global scale [11]. Research has indicated that the inhibition of mitochondrial complex III via AZOX effectively disrupts the mitochondrial respiratory chain, resulting in a decline in ATP synthesis within fungal cells [12]. Consequently, this impedes mycelial growth and serves as a preventive measure against the development and proliferation of pathogenic spores [13]. It is able to block the development of ascospores that can be considered as an indirect way to control *S. sclerotiorum* spreading, since sclerotia, which provides the development of the apothecium and ascospores, may initiate new disease development cycles, thus complicating the control of this pathogen [14]. It can also be used to control *Ascochyta*, *Powdery mildew*, and other diseases transmitted through the foliar and soil [15]. The limited water solubility of AZOX (6.7 μg/mL), and it is characterized by poor solubility, usually results in low bioavailability [16,17,18]. In addition, AZOX remaining in the environment may negatively affect the physiological activities of fish, algae, and earthworms [19,20,21]. Hence, the carrier capable of precisely transporting AZOX fungicide to the designated location within the system and effectively releasing it holds significant scientific and practical significance in enhancing its bioavailability [22].

The zeolitic imidazolate framework materials (ZIFs), such as metal–organic framework materials (MOFs), have demonstrated promising potential in the field of adsorption for organic compounds due to their periodic network structure, tunable pore size, high specific surface area, and exceptional stability within molecular sieves [23]. One of the subclasses of MOF is known as zeolitic imidazolate framework-8 (ZIF-8), which is produced through the synthesis of Zn^2+^ and 2-methylimidazole (H-MelM) [24]. Due to its rapid release in acidic environments, low toxicity, and excellent biocompatibility, this substance exhibits significant potential for facilitating the delivery of biomacromolecular medications [25]. Compared to other MOFs incorporating heavy metals, Zn^2+^ demonstrates superior environmental compatibility. Moreover, it not only effectively mitigates tomato lobular disease, but also actively participates in enzymatic catalytic processes within the human body [26].The growth and reproduction of *Botrytis cinerea* also leads to the release of large amounts of oxalic acid, which can lead to the disintegration of ZIF-8 [27]. Therefore, ZIF-8 offers an opportunity for constructing a pH-responsive drug delivery system to achieve intelligent fungicide release in agriculture [28]. However, conventional techniques commonly lead to agglomeration of ZIF-8 particles, resulting in a reduction of adsorption sites and specific surface area. Hence, utilizing biochar as a matrix for ZIF-8 not only alleviates particle agglomeration due to its abundant functional groups, but also enhances specific surface area [29]. The presence of numerous oxygen-containing functional groups on BC effectively mitigates aggregation of ZIF-8 and enhances dispersion of Zn^2+^ [30]. 

In this study, AZOX was adsorbed onto ZIF-8/BC by impregnation method, and loaded AZOX was released in response to oxalic acid produced by *Botrytis cinerea* for *Botrytis cinerea* control (Figure 1). The efficacy of AZOX-ZIF-8/BC nanoparticles in controlling *Botrytis cinerea* was investigated through both in vitro and pot experiments. These nanoparticles not only enhance the efficiency of AZOX, but also demonstrate promising potential as an intelligent disease control nanoplatform.

## 2. Results and Discussion

### 2.1. Thermal Gravimetric Analysis

#### 2.1.1. ZIF-8/BC Fabrication

In this investigation, AZOX was impregnated into ZIF-8/BC nanoparticles via a 25 °C method to synthesize AZOX-ZIF-8/BC. ZIF-8 particles were prepared using three distinct solvents: methanol, deionized water, and ammonia. The resulting sizes of ZIF-8 (C), ZIF-8 (H), and ZIF-8 (N) particles were measured as 145 nm, 893 nm, and 4160 nm, respectively (Figure 2a–c). The absorption of drugs by plants could be influenced by the particle sizes of carriers, with the size of particles in the carrier directly impacting drug load and inversely affecting release efficiency [31]. The surface characteristics and particle size of ZIF-8 (N) might impede efficient drug release, whereas ZIF-8 (H) exhibits a cubic-like morphology and ZIF-8 (C) assumes a rhombic dodecahedral shape [32,33]. Due to their reduced surface energy, rhombic dodecahedron microcrystals exhibit enhanced stability compared to cubes [34]. Furthermore, the uniform dispersion of ZIF-8 (C) particles was more pronounced. Therefore, methanol was employed as a solvent in this investigation for subsequent synthesis of ZIF-8.

The morphology of ZIF-8/BC nanoparticles was illustrated in Figure 2d, demonstrating the successful in-situ growth of ZIF-8 nanoparticles on biochar. This confirmed the effective integration of ZIF-8 and biochar, showcasing a regular rhombic dodecahedral structure with uniform particle size and smooth appearance for ZIF-8. A comparison with Figure 2a revealed a reduction in agglomeration phenomenon upon combining ZIF-8 with biochar.

#### 2.1.2. AZOX Loading

By optimizing the mass ratio of AZOX to carrier, we determined the optimal conditions for loading capacity (LC) and encapsulation efficiency (EE), as presented in Table 1. The loading content was directly proportional to the concentration of AZOX, potentially attributed to its ability to enhance drug molecule adsorption on the carrier. We achieved a maximum load content of AZOX at 19.83 ± 0.25% with a mass ratio of AZOX to carrier set at 3:1. However, when maintaining constant carrier weight, an inverse relationship between the concentration of AZOX and the encapsulation rate was observed. Therefore, considering both LC and EE, we subsequently prepared AZOX-ZIF-8/BC using a 3:1 ratio of AZOX to carrier for characterization, release analysis, and bioactivity determination.

### 2.2. Sample Characterization

#### 2.2.1. FT-IR Spectra

The FT-IR spectra of BC, ZIF-8/BC, AZOX, and AZOX-ZIF-8/BC are depicted in Figure 3a. In the spectrum of ZIF-8/BC, the absorption peak at 3138 cm^−1^ corresponds to the stretching vibration of aromatic C=H bonds within the imidazole ring. Moreover, the absorption peak at 420 cm^−1^ was attributed to the stretching motion of Zn-N bonds, which was evident in the curve for AZOX-ZIF-8/BC. Furthermore, distinctive absorption peaks at 1561 and 2231 cm^−1^ were observed in AZOX alone and could also be discerned in AZOX-ZIF-8/BC. These findings indicated successful encapsulation of AZOX into ZIF-8/BC.

#### 2.2.2. Thermal Gravimetric Analysis

The thermal stability and decomposition behavior of materials were investigated through thermogravimetric analysis (TGA). Figure 3b,c presents the TGA and DTG curves for AZOX, ZIF-8/BC, and AZOX-ZIF-8/BC. A significant weight loss was observed for AZOX in the temperature range of 280 to 380 °C, with the maximum thermal decomposition rate occurring at 340 °C. In contrast, AZOX-ZIF-8/BC exhibited rapid weight loss within the temperature range of 330 to 420 °C, with the highest loss recorded at 380 °C. This phenomenon could be attributed to the load on the crystal structure, which shifts the initial decomposition temperature of AZOX towards higher values when combined with ZIF-8/BC. These findings suggested that ZIF-8/BC demonstrates excellent thermal stability for AZOX and has potential to extend its lifespan in practical applications.

#### 2.2.3. XRD Analysis

The characterization results of AZOX-ZIF-8/BC and ZIF-8/BC are shown in Figure 4. The XRD pattern shows that the seven diffraction peaks at 7.31, 10.31, 12.71, 14.71, 16.41, 18.01, and 26.71 can point to (011), (002), (112), (022), (013), (222), and (134) crystal planes, respectively [35]. The diffraction peaks and their positions observed in this study are consistent with previous literature findings, providing further evidence of the presence of ZIF-8 within the sample. The similarity in characteristic peak between AZOX-ZIF-8/BC and ZIF-8/BC suggests that the crystal structure of ZIF-8 remains unaffected by AZOX during the loading process, thereby maintaining its integrity. Furthermore, a decrease in strength value for AZOX-ZIF-8/BC indicates successful encapsulation of AZOX molecules within the pores of ZIF-8.

#### 2.2.4. Adsorption/Desorption Isotherms and Micropore Size

The microstructural analysis of inorganic porous materials involved conducting Nitrogen adsorption–desorption experiments at 77.3 K to determine the specific surface area, pore volume, and pore size of the prepared ZIF-8/BC composite. The results obtained, as depicted in Figure 5a,b, indicated a rapid increase in nitrogen adsorption capacity for dodecahedron crystals when p/p0 was less than 0.2. This behavior exhibited Langmuir adsorption isotherm characteristics of type I. Furthermore, the pore size distribution curve confirmed the presence of typical microporous adsorption. The measured values for BET specific surface area (SBET), BJH pore size (DBJH), and total pore volume (Vt) were determined to be 1261.57 m^2^/g, 14.481 cm^3^/g, and 4.59 nm, respectively.

### 2.3. AZOX Release Behavior

Controlling the release of fungicide in response to stimuli was crucial for reducing fungicide usage and promoting agricultural development. The rate of release can provide valuable insights into potential strategies for regulating the release process. Under acidic conditions with a pH of 4.8, a significantly higher release rate was observed compared to neutral and alkaline conditions. At 55 h and pH levels of 4.8, 7.0, and 8.2, respectively (Figure 6a), the percentages of AZOX released were found to be 76.60%, 58.56%, and 51.61%. This remarkable ability of AZOX-ZIF-8/BC to respond to changes in pH can be attributed to its susceptibility towards protonic acid attack, which leads to the disruption of Zn-N bonds in ZIF-8, resulting in rapid collapse of its crystal structure, thereby facilitating significant release of AZOX molecules.

In order to gain a comprehensive understanding of the release mechanism of AZOX, we employed various mathematical models including zero-order, first-order, and Korsmeyer–Peppas models to analyze the release kinetics. These mathematical models are widely used in describing drug release from polymer systems. The fitting curves for these mathematical models under different pH conditions can be observed in Figure 6b–d as well as Table 2. At pH levels of 7.0 and 8.2, the experimental data of AZOX exhibited excellent agreement with the first-order model. By utilizing the Korsmeyer–Peppas model, we successfully predicted the release kinetics of AZOX-ZIF-8/BC nanoparticles at a pH level of 4.5. It can be seen that the AZOX release of AZOX-ZIF-8/BC nanoparticles is mainly affected by Fick diffusion (*n* ≤ 4.5) [36].

### 2.4. In Vitro Fungicidal Activity 

In order to explore the prevention and treatment effect of AZOX-ZIF-8/BC on *Botrytis cinerea*, the effect of different concentrations of AZOX-ZIF-8/BC on *Botrytis cinerea* mycelial growth rate was studied by mycelial growth rate method. Meanwhile, vector materials ZIF-8/BC and AZOX were used as control group. Under the conditions of this experiment, after inoculation with *Botrytis cinerea* 7 days, the diameter of the blank group was 68 ± 2.87 mm, and the diameter of the hypha of AZOX-ZIF-8/BC, AZOX, and ZIF-8/BC were all reduced. As the concentration of the three substances increased, the growth of *Botrytis cinerea* was significantly affected. It can be observed that the carrier ZIF-8/BC also had an effect on the growth of *Botrytis cinerea* (Figure 7).

It can be seen from Figure 8 that AZOX-ZIF-8/BC had the best control effect under the same concentration condition. At the same concentration, the antibacterial activity of AZOX-ZIF-8/BC was 41.15%, 43.13%, 20.84%, and 51.37% higher than that of AZOX from 40 to 5 mg/L, respectively. Therefore, the results showed that the presence of vector ZIF-8/BC could effectively improve the control effect of AZOX and inhibit the growth and penetration of mycelia. This is because the oxalic acid produced during the growth of *Botrytis cinerea* leads to the breakdown of ZIF-8, resulting in the release of Zn^2+^, thus hindering the growth of mycelia. Moreover, the disintegration of ZIF-8 promoted the rapid release of AZOX, enhancing its effectiveness.

### 2.5. Study on the Fungicidal Activity of Potted Plants

The control effect of AZOX-ZIF-8/BC on *Botrytis cinerea* was further verified through tomato pot experiment (Figure 9a). As shown in the figure, when AZOX concentration was 500 and 400 mg/L, tomato leaves were healthy, indicating that AZOX-ZIF-8/BC and AZOX control group could play a good control effect. When AZOX concentration was 300, 200, and 100 mg/L, the number and size of lesions on AZOX-ZIF-8/BC were significantly smaller than those in the control group. AZOX-ZIF-8/BC had better control effect than AZOX. When AZOX concentration was 0 mg/L, tomato plants were in poor health or even about to wither, indicating that *Botrytis cinerea* will cause devastating damage to tomatoes.

In order to express the effect of pot experiment more clearly, the disease index and control effect were used to explain. As shown in Figure 9b,c, there was no statistically significant difference between 500 and 400 mg/L AZOX and 500 and 400 mg/L AZOX. This might be due to the sufficient use of AZOX, but it might also reduce the efficiency of AZOX and lead to higher fungicide residues. When AZOX concentration was 300 mg/L, the control effect of AZOX-ZIF-8/BC was significantly better than that of AZOX control group, which could be attributed to its ZIF-8/BC carrier, which could effectively prevent the premature degradation of AZOX and achieve accurate release. When the concentration of AZOX was 200 and 100 mg/L, the control effect of AZOX-ZIF-8/BC was better than that of the control group, but the residual *Botrytis cinerea* might infect tomato plants again on a large scale, which could be because the concentration of AZOX was too low to achieve the ideal control effect. Therefore, 300 mg/L concentration of AZOX-ZIF-8-BC can be considered as a safe and effective control of *Botrytis cinerea*.

## 3. Materials and Methods

### 3.1. Materials

2-Methylimidazole (C_4_H_6_N_2_), dichloromethane (CH_2_Cl_2_), formic acid (HCOOH), Tween 80, and AZOX were purchased from McLean (Shanghai, China); zinc nitrate hexahydrate (Zn(NO_3_)_2_·6H_2_O) and acetonitrile were purchased from Aladdin Industries (Shanghai, China); ammonium hydroxide (NH_4_OH) was purchased from Beijing Chemical Plant(Beijing, China). Methanol and ethanol were purchased from Guoyao Group Chemical Reagent Co., Ltd. (Shanghai, China); potato Glucose Agar was purchased from National Pharmaceutical Group Chemical Reagents Co., Ltd (Shanghai, China). 

*Botrytis cinerea*, isolated from the pathogen-infected tomato plants, was kindly provided by Laboratory of Pesticide Bioassay, Plant Protection College at Jilin Agricultural University.

### 3.2. Thermal Gravimetric Analysis

#### 3.2.1. Synthesis of ZIF-8

The preparation procedure for ZIF-8 was as follows: 0.372 g of Zn(NO_3_)_2_·6H_2_O was dissolved in 25 mL of methanol at 25 °C, followed by the addition of a methanol solution containing 0.821 g of H-MelM (25 mL) under magnetic stirring. The mixture was vigorously stirred in a thermostatic water bath at 600 rpm and left at 25 °C for 24 h. After that, it was subjected to centrifugation at a speed of 8000 rpm for 15 min. The resulting centrifuged product was collected and subsequently washed three times with deionized water and anhydrous ethanol. The product was subjected to a drying process in an oven at 60 °C for 12 h, ultimately resulting in its designation as ZIF-8 (C). Similarly, ZIF-8 (H) was synthesized using deionized water as the solvent following the identical procedure. Another variant, denoted as ZIF-8 (N), was prepared utilizing ammonia as the solvent [37].

#### 3.2.2. Preparation of BC

The corn straw was subjected to pyrolysis at a temperature of 450 °C for one hour in a fixed bed under a nitrogen atmosphere (flow rate: 300 mL/min) to produce biochar, which was subsequently designated as ‘BC’ [30].

#### 3.2.3. Preparation of AZOX Loaded ZIF-8/BC

To initiate the experiment, the synthesis of ZIF-8/BC nanoparticles was carried out. The dissolution process involved combining 0.372 g of Zn(NO_3_)_2_·6H_2_O and 25 mg of BC and 0.821 g of H-MelM in separate containers containing 25 mL of methanol at ambient temperature. After a 10-min period of magnetic stirring, a solution containing H-MelM in methanol (25 mL) was gradually added dropwise, followed by continuous stirring at 25 °C for a duration of 24 h. After the reaction, the precipitate was separated via centrifugation at a speed of 8000 rpm for a duration of 15 min. The resulting centrifuged product was collected and subsequently washed three times with deionized water and anhydrous ethanol. To obtain ZIF-8/BC nanoparticles, the harvested precipitate was then subjected to drying in an oven set at a temperature of 60 °C for a period of 12 h [38].

The next step involved preparing a solution of AZOX by dissolving 600 mg of AZOX in 30 mL of dichloromethane, resulting in a concentration of 20 mg/mL. The subsequent step involved introducing a mixture of ZIF-8/BC nanoparticles and varying ratios of fungicide/carrier into the aforementioned solution. This resulting mixture was then sealed and subjected to magnetic stirring at 25 °C for 24 h, leading to the formation of AZOX loaded ZIF-8/BC nanoparticles [39]. Following centrifugation of the mixture at a speed of 8000 revolutions per minute for a duration of 15 min, the resulting solid was collected and subjected to three consecutive washes using 100 mL methanol. Subsequently, freeze-drying was employed to eliminate any remaining moisture from the collected material. The synthesized compound was designated as AZOX-ZIF-8/BC.

### 3.3. Characterization

The morphology and structural characteristics of the prepared nanoparticles were analyzed by scanning electron microscopy (SEM, SS-550, Shimadzu, Japan).

Fourier transform infrared spectroscopy was used to record in the spectral region of 400 to 4000 cm^−1^ on a spectrometer (FT-IR, IRAffinity-1S, Shimadzu, Japan).

Thermogravimetric analysis was performed using a thermogravimetric analyzer (TGA, HCT-3, Beijing Hengjiu, China), operated at an N_2_ flow rate of 30 mL min^−1^ and a heating rate of 10 °C min^−1^ at temperatures from 30 to 800 °C. 

The crystallinity of ZIF-8/BC and AZOX-ZIF-8/BC was analyzed by XRD using D8 ADVANCE in a scanning range of 5–50° at a rate of 2° min^−1^.

The specific surface area and pore size distribution were examined using a specific surface area and pore size analyzer (TriStarII 3020, Micromeritics Instruments Corp, Norcross, GA, USA) at 200 °C for nitrogen adsorption/desorption measurements.

### 3.4. Evaluation of ZIF-8/BC Nanoparticle Loading Content

The loading content and encapsulation efficiency of AZOX-ZIF-8/BC were evaluated by dispersing a 10 mg sample in 10 mL of methanol. The dispersion underwent ultrasonic treatment for 1 h, followed by centrifugation at 8000 rpm for 15 min and filtration through a filter with a pore size of 0.22 μm. Analysis of AZOX concentration was conducted using high-performance liquid chromatography (HPLC) equipped with a diode array detector on an Agilent 1200 Infinity series instrument from Agilent Technologies Inc., Santa Clara, CA, USA. The mobile phase consisted of acetonitrile/0.1% formic acid aqueous solution (*V*/*V* = 48:52), flowing at a rate of 1.0 mL min^−1^ through an Agilent Technologies Inc., Santa Clara, CA, USA C18 reversed-phase column (2.1 × 100 mm, inner diameter: 2.7 mm). Detection wavelength was set at 254 nm, injection volume at 20 μL, analysis performed at 25 °C, and measurements repeated three times.

The loading amount and encapsulation efficiency were calculated as follows: loading content (%) = (weight of AZOX in AZOX-ZIF-8/BC/weight of AZOX-ZIF-8/BC) × 100%; packaging efficiency (%) = (AZOX weight in AZOX-ZIF-8/BC/initial weight of AZOX) × 100%.

### 3.5. In Vitro Release Behavior

We studied the release behavior of AZOX-ZIF-8/BC at three pH values (4.8, 7.0, 8.2). Considering the low water solubility of AZOX, we used a mixed solution of PBS, ethanol, and Tween-80 emulsifier (70:29.5:0.5, *v*/*v*/*v*) as the release medium. The 20 mg AZOX-ZIF-8/BC was dispersed in a 2.0 mL release medium (molecular weight cut-off: 8000–14,000 Da) in a dialysis bag. The sealed dialysis bag was then placed in a 200 mL release medium at 30 °C and stirred at 100 rpm using a D-800LS dissolution tester. The 1.0 mL release medium was taken out every certain time for HPLC analysis, and an equal volume of fresh release medium was added. Determine the process in three parts. The cumulative release of AZOX was calculated according to the following formula:$$E_{r}=\frac{V_{e}\sum_{i=0}^{n-1}C_{i}+V_{o}C_{n}}{m_{fungicide}}\times 100\%$$

*E_r_* is the cumulative AZOX (%) released by AZOX-ZIF-8/BC; *V_e_* = 1.0 mL; *C_n_* is the AZOX concentration (mg/mL) in the release medium at the sampling time (*n*); *V_o_* was the volume of the release solution (200 mL); *m_fungicide_* is the total amount of fungicide loaded into ZIF-8/BC (mg).

### 3.6. In Vitro Fungicidal Activity

The test used the ‘Pesticide Indoor Bioassay Test Criteria Fungicides Part II: Inhibition of Pathogenic Fungal Mycelial Growth Test Plate Method’ (NY/T 1156.2-2006) [40]. The samples and AZOX were diluted with 0.1% Tween 80 aqueous solution to different series of mass concentrations, and PDA was treated to make 40, 30, 20, 10 mg/L drug-containing plates. The treatment without medicament was set as blank control, and each treatment was repeated three times. The 5 mm punch was used to cut the fungus cake at the edge of the cultured pathogen colony and inoculate it in the center of the plate. The mycelium was face up and placed in the incubator. After 7 days of culture, the colony diameter was measured by the cross method, and the average value was taken as mm. The calculation formula is as follows:$$I=\frac{D_{o}-D_{t}}{D_{o}}\times 100\%$$
where *I* is the inhibition rate of mycelial growth; *D_o_* is the growth diameter of the blank control colony; *D_t_* is the colony growth diameter of the drug treatment.

### 3.7. Study on the Fungicidal Activity of Potted Plants

The test used the ‘Pesticide Indoor Bioassay Test Criteria Fungicides Part 10: Prevention of Gray Mold Test Pot Method’ (NY/T 1156.10-2008). After cultivating the pathogen to cultivate spores, a suspension of 1 × 10^5^ spores/mL was prepared for future use by rinsing the spores with sterile water and filtering them through two layers of gauze. The sample and AZOX were diluted with a 0.1% aqueous solution of Tween 80 to create five different concentrations. The liquid was evenly sprayed onto the entire leaf surface until it naturally dried. Each treatment consisted of three pots with four replicates, and a control blank was included for comparison purposes. Inoculation was performed by spraying the spore suspension inoculation solution; for the protective test, inoculation took place 24 h after treatment, while for the therapeutic test, inoculation occurred prior to administering the medication. After inoculation, they were transferred to a humid chamber with a relative humidity above 95%, maintained at a temperature range of 20 °C to 22 °C in complete darkness for 24 h. Subsequently, they were cultured under alternating light (12 h) and dark (12 h) conditions at temperatures ranging from 20 °C to 25 °C with a relative humidity between 80% and 90% for 7 days.

When the proportion of unhealthy foliage in the control group exceeded 50%, an assessment was conducted to determine the prevalence of each treatment, with a minimum of 30 leaves examined for each treatment. The grading method was as follows: level zero indicates the absence of disease spots [41]. Level one corresponds to lesions covering less than five percent of the entire leaf surface. For level three, lesions account for between five and fifteen percent of the overall leaf area. Moving up to level five, we observed lesions occupying fifteen to twenty-five percent of the total leaf surface. At level seven, lesions extend from twenty-five to fifty percent coverage on leaves. Finally, at level nine, lesions encompass over fifty percent of a given leaf’s surface area. Based on collected survey data, calculations were conducted to determine both disease index and treatment effectiveness. The formula is as follows:$$X=\frac{\sum \left(N_{V}\times V\right)}{N\times 9}\times 100$$
where *X*—disease index; *N_v_*—disease leaf index at all levels; *v*—Relative Grade Index; *n*—Total number of leaves surveyed.
$$P=\frac{CK-PT}{CK}\times 100$$
where *P*—control effect, the unit is percentage (%); *CK*—blank control disease index; *PT*—drug treatment disease index.

## 4. Conclusions

In this research, we developed a carrier called ZIF-8/BC that responds to changes in pH. The purpose was to load AZOX for the prevention and treatment of *Botrytis cinerea*. Through XRD, FT-IR, and TGA analysis, it was observed that AZOX could be easily absorbed into ZIF-8/BC using a simple impregnation technique. SEM and N_2_ adsorption/desorption data confirmed that ZIF-8/BC possesses typical microporous adsorption properties and effectively loads AZOX. Release testing demonstrated the pH-responsive behavior of ZIF-8/BC towards AZOX; specifically, at a pH of 4.8, an initial burst release followed by sustained release occurred. This unique characteristic not only ensures rapid fungal eradication upon application, but also provides long-lasting control effects. In vitro fungicidal experiments revealed that the growth of *Botrytis cinerea* triggers the disintegration of AZOX-ZIF-8/BC, resulting in the release of both AZOX and Zn^2+^. Furthermore, pot experiments demonstrated the excellent efficacy of AZOX-ZIF-8/BC against *Botrytis cinerea* while significantly reducing the amount of applied fungicide required. Consequently, our development of pH-responsive carriers presents an innovative approach to enhance fungicide efficiency while minimizing usage—a strategy with promising applications in future agricultural practices.

## Data Availability

Not applicable.

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
