# Peer review of "Preparation of Azoxystrobin-Zinc Metal–Organic Framework/Biomass Charcoal Composite Materials and Application in the Prevention and Control of Gray Mold in Tomato"

_ijms, 2023, doi:10.3390/ijms242115609_

Round 1

Reviewer 1 Report

Comments:

To the Editor Prof.

The manuscript “Preparation of azoxystrobin-zinc metal-organic framework/biomass charcoal composite materials and application in the prevention and control of gray mold in tomato” describes the efficiency of azoxystrobin-zinc metal-organic framework/biomass charcoal composite materials against Botrytis cinerea, the cause of tomato gray mold disease. The study is interesting and of great practical importance. But major revisions are suggested to make the article satisfies the criteria of the International Journal of Molecular Sciences.

To the Authors

1. the abstract and introduction could benefit from providing more context about the importance and impact of gray mold disease in tomato. This would help readers understand the significance of the research and its potential implications.

2. Improve formatting of the paper especially space between words.

3. Arrange keywords alphabetically.

4. Mention the name of Botrytis cinerea italic throughout the manuscript.

5. Mention the sourc of Botrytis cinerea in Materials and Methods.

Dear reviewer:

Thank you very much for your professional comments on our articles. As you are concerned, there are several problems to be solved. According to your suggestions, we have made many changes to the previous manuscript. We will provide you with two documents, one of which is marked in pink after modification according to your question. On the other hand, based on your comments, I would like to make the following general reply:

Question 1:The abstract and introduction could benefit from providing more context about the importance and impact of gray mold disease in tomato. This would help readers understand the significance of the research and its potential implications.

Reply 1:Thank you for pointing this out. We agree with this comment.

The abstract section has been revised in line 19, the first paragraph of the introduction has been changed from the original five references to 10, and provides the reader with more detailed information from various perspectives.

Question 2:Improve formatting of the paper especially space between words.

Reply 2:We are sorry for our carelessness. In our resubmitted manuscript, the format has been revised. Thank you for your correction.

Question 3:Arrange keywords alphabetically.

Reply 3:Thank you for your careful examination. We've arranged the keywords in alphabetical order.

Question 4: Mention the name of Botrytis cinerea italic throughout the manuscript.

Reply 4:Once again, we are sorry for our carelessness. In our resubmitted manuscript, Botrytis cinerea has been modified. Thank you for your correction.

Question 5:Mention the sourc of Botrytis cinerea in Materials and Methods.

Reply 5: Botrytis cinerea produces oxalic acid as it grows.Explanation is given on page 2, line 78, and references are cited [27].

Reviewer 2 Report

The manuscript treats an important and recent issue in the field of plant protection; however, it still needs a substantial revision. I will summarize the comments in the following:

General comments:

The whole manuscript needs to be reedited, you need to delete a space in many cases, and you need to add a space in other cases, please refer to the highlighted words or phrases on the PDF file.

Please use a uniformed style of the reference citation in the text, you use in bracts style [] , however sometimes you use subscripted number of references.

Use uniform units throughout the manuscripts, you use 2 different styles.

Add a follow stop after citation not before it.

Please write the species names in italic font "Botrytis cinerea "

All highlighted parts on the PDF needs to be changed.

Specific comments:

In materials and methods section, you must give some details about the pathogen " Botrytis cinerea " that you have used for example where did you get it, isolate number, source of isolation. You neglected it however it is the key of your work.

I think the term "pesticide" should be changed to  "fungicide" because it is a specific term.  

Throughout the manuscript, please put the figure or the table after the text not before.

In section  "3.9." change " Study on the bactericidal activity of potted plants" to Study on the fungicidal activity of potted plants, and through the text.

In figure 9 (a), I can't see any change in the treatments, you may substitute it with more clear photo or add some thing to distinguish the difference such as adding a scale !!   

English showed be revised and edited before publication 

Author Response

Dear reviewer:

Thank you very much for your professional comments on our articles. As you are concerned, there are several problems to be solved. According to your suggestions, we have made many changes to the previous manuscript. We will provide you with two documents, one of which is marked in pink after modification according to your question. On the other hand, based on your comments, I would like to make the following general reply:

Question 1:The whole manuscript needs to be reedited, you need to delete a space in many cases, and you need to add a space in other cases, please refer to the highlighted words or phrases on the PDF file.

Reply 1:We were really sorry for our careless mistakes. Thank you for your reminder.

We have carefully read and revised the content of the manuscript, every sentence and every word. And refer to the highlighted word or phrase on the PDF file.

Question 2:Please use a uniformed style of the reference citation in the text, you use in bracts style [] , however sometimes you use subscripted number of references.

Reply 2:We apologize for our carelessness. In our resubmitted manuscript, the figures for the references have been revised. Thank you for your correction.

Question 3:Use uniform units throughout the manuscripts, you use 2 different styles.

Reply 3:Thank you for your careful reading, we revised the manuscript and used a unified unit, and carefully read the article again.

Question 4:Add a follow stop after citation not before it.

Reply 4:We have revised the quotation, replacing the original “.[]” with “[].”.

Question 5:Please write the species names in italic font "Botrytis cinerea ".All highlighted parts on the PDF needs to be changed.

Reply 5:We modified the word or phrase highlighted on the PDF file, especially the Botrytis cinerea was also selected in italics.

Question 6:Specific comments:

In materials and methods section, you must give some details about the pathogen " Botrytis cinerea " that you have used for example where did you get it, isolate number, source of isolation. You neglected it however it is the key of your work.

Reply 6:Botrytis cinerea was isolated and identified on infected tomatoes, but no information such as number could be provided. The details are on page three, lines 104 through 106.

Question 7:I think the term "pesticide" should be changed to  "fungicide" because it is a specific term.

Reply 7:I feel deeply guilty for my poor English. “pesticide” should not be written by me in the article, “fungicide” is the most correct way to write. Thank you very much for your reminding.

Question 8:Throughout the manuscript, please put the figure or the table after the text not before.

Reply 8:Thanks for your suggestion, we put the graph or table after the text.

Question 9:In section  "3.9." change " Study on the bactericidal activity of potted plants" to Study on the fungicidal activity of potted plants, and through the text.

Reply 9:Thank you for your suggestion. “Study on the fungicidal activity of potted plants” is the most correct way to write it. After correction, we have modified similar contents in the whole paper.

Question 10: In figure 9 (a), I can't see any change in the treatments, you may substitute it with more clear photo or add some thing to distinguish the difference such as adding a scale !! 

Reply 10:There are two types of in vitro experiments: leaf method and pot method. The treatment effect of the leaf method is better and more obvious, and it is also a commonly used method. The potting method is not clearly expressed in the photo, so it is used less frequently, but is closer to the actual needs of the planting. The potting method is strictly implemented in accordance with Chinese national standards (NY/T 1156.10-2008), and the data is authentic and reliable. In order to better answer everyone's doubts, the text content of line 3, 5.11 on page 377 is explained in more detail, and more data is added to show the prevention effect.

Reviewer 3 Report

The article entitled “Preparation of Azoxystrobin-Zinc metal-organic framework/ Biomass

charcoal composite materials and Application in the prevention and control of gray mold in tomato” describes the pH-responsive pesticide delivery system using Zinc metal-organic framework/Biomass charcoal (ZIF-8/BC). The authors were used Azoxystrobin pesticide and prepared AZOX-ZIF-8/BC nanopesticides by impregnation method. In vitro, and pot experiments show excellent efficacy of AZOX-ZIF-8/BC against Botrytis cinerea.

 I recommend it is highly suitable for publication in International Journal of Molecular Sciences after minor revision. The revisions are,

Throughout the manuscript the authors should use ambient temperature and the Botrytis cinerea should be in italics throughout the manuscript.

Results and Discussion

Page number 7, Table 1. The authors should correct the spelling to Loading Content (%) instead of Lading Content (%).

 Conclusion

Page number 11, line number 374, the authors should change the N2 instead of N2

Author Response

Dear reviewer:

Thank you very much for your professional comments on our articles. As you are concerned, there are several problems to be solved. According to your suggestions, we have made many changes to the previous manuscript. We will provide you with two documents, one of which is marked in pink after modification according to your question. On the other hand, based on your comments, I would like to make the following general reply:

Question 1:Throughout the manuscript the authors should use ambient temperature and the Botrytis cinerea should be in italics throughout the manuscript.

Reply 1:Thanks for your comments, we changed the room temperature in the article to 25 ° C and referred to the words and phrases highlighted on the PDF file, in particular correcting the Botrytis cinerea to italics.

Question 2:Results and Discussion

Page number 7, Table 1. The authors should correct the spelling to Loading Content (%) instead of Lading Content (%).

Reply 2:We sincerely thank the reviewer for careful reading. As suggested by the reviewer, change to“Loading”.

Question 3:Page number 11, line number 374, the authors should change the N2 instead of N2.

Reply 3:We sincerely thank the reviewer for careful reading. As suggested by the reviewer, change to“N2”.

Reviewer 4 Report

The work concerns an important aspect of plant cultivation. The authors focused on developing an effective method for the absorption of biologically active AZOX. This compound has documented activity as a pesticide. Designing safe, highly active pesticides is a major challenge. The authors managed to achieve their goal by using the specifically designed Zinc metal-organic frameworks. The work and the results obtained are of great practical importance. The authors managed to obtain a preparation with the desired characteristics that allows for controlled dosing of the pesticide. The experiments are well planned and the results obtained are well interpreted. The authors used a number of analytical methods to fully characterize the obtained MOFs. The work deserves to be published in its current form.

Author Response

Dear reviewer:

Thank you so much for your comments.

Reviewer 5 Report

Comments/suggestions to the authors:

Introduction should be improved.

Page 1, line 33: Botrytis cinerea should be written in italics (and throughout the whole manuscript as well). Similarly, other phrases like in vitro should follow the same rule. 

Page 1, line 37: What is “Revised text:” stands for in the manuscript?

Page 1, line 41: What AZOX stands for? Before the use of acronyms authors should mention the proper name and put the acronym in parenthesis.

Page 1, line 4: It has been a number of lead molecules that derived from a number of basidiomycetes, such as Strobilurus tenacellus, that were then developed to azoxystrobin, as we use it today, and other important fungicides and oomycetides. Thus, authors should develop more in details on this, since azoxystrobin plays a pivotal role in their research. 

Page 2, lines 47-48: “The limited water solubility of AZOX (6.7μg/mL). [8] However, has significantly impeded its application in plant protection.

To the best of my knowledge azoxystrobin has been the pioneer strobilurin fungicide and has been registered in numerous crops against major plant pathogens worldwide for over 20 years. Thus, I cannot accept the aforementioned statement of its impended application, especially if it is not supported by any relevant references, at least. 

Page 2, line 75: What BF stands for?

Page 5, line 193: 2.8. Study on the bactericidal activity of potted plants”.

Botrytis cinerea is not a bacterium, but an ascomycete, a true fungus. Thus, the term bactericidal is not proper and should be replaced by the term fungicidal. 

Page 10, paragraph 3.8: Authors present results regarding the in vitro activity on the mycelial growth od B. cinerea. However, their analysis is only qualitative, missing numerical, quantitative data and statistical analysis. 

Page 11, paragraph 3.9:

(1) In Fig. 9b azoxystrobin performs better to AZOX-ZIF-8/BC in terms of disease index, biologically meaning that with azoxystrobin we estimated less disease level compared to AZOX-ZIF-8/BC. They are statistically not significantly different only at the concentration of 500 ppm. This outcome is not clearly mentioned in the text of 3.9. 

(2) It is not clear the reverse of Fig 9b when interpreting the same data to Fig 9c at the concentrations of 100, 200, and 300 ppm. Please provide sufficient biological explanation.

(e) Authors in the text of 3.9 as well as in Fig. 9 present data without mentioning whether these data refer to preventative or therapeutic application of the active ingredient. In Paragraph 2.8 they write that they performed the experiment using both application regimes. 

Overall, the text has numerous syntax and grammatic errors that need to be taken care of. 

Comments/suggestions to the authors:

Introduction should be improved.

Page 1, line 33: Botrytis cinerea should be written in italics (and throughout the whole manuscript as well). Similarly, other phrases like in vitro should follow the same rule. 

Page 1, line 37: What is “Revised text:” stands for in the manuscript?

Page 1, line 41: What AZOX stands for? Before the use of acronyms authors should mention the proper name and put the acronym in parenthesis.

Page 1, line 4: It has been a number of lead molecules that derived from a number of basidiomycetes, such as Strobilurus tenacellus, that were then developed to azoxystrobin, as we use it today, and other important fungicides and oomycetides. Thus, authors should develop more in details on this, since azoxystrobin plays a pivotal role in their research. 

Page 2, lines 47-48: “The limited water solubility of AZOX (6.7μg/mL). [8] However, has significantly impeded its application in plant protection.

To the best of my knowledge azoxystrobin has been the pioneer strobilurin fungicide and has been registered in numerous crops against major plant pathogens worldwide for over 20 years. Thus, I cannot accept the aforementioned statement of its impended application, especially if it is not supported by any relevant references, at least. 

Page 2, line 75: What BF stands for?

Page 5, line 193: 2.8. Study on the bactericidal activity of potted plants”.

Botrytis cinerea is not a bacterium, but an ascomycete, a true fungus. Thus, the term bactericidal is not proper and should be replaced by the term fungicidal. 

Page 10, paragraph 3.8: Authors present results regarding the in vitro activity on the mycelial growth od B. cinerea. However, their analysis is only qualitative, missing numerical, quantitative data and statistical analysis. 

Page 11, paragraph 3.9:

(1) In Fig. 9b azoxystrobin performs better to AZOX-ZIF-8/BC in terms of disease index, biologically meaning that with azoxystrobin we estimated less disease level compared to AZOX-ZIF-8/BC. They are statistically not significantly different only at the concentration of 500 ppm. This outcome is not clearly mentioned in the text of 3.9. 

(2) It is not clear the reverse of Fig 9b when interpreting the same data to Fig 9c at the concentrations of 100, 200, and 300 ppm. Please provide sufficient biological explanation.

(e) Authors in the text of 3.9 as well as in Fig. 9 present data without mentioning whether these data refer to preventative or therapeutic application of the active ingredient. In Paragraph 2.8 they write that they performed the experiment using both application regimes. 

Overall, the text has numerous syntax and grammatic errors that need to be taken care of. 

Author Response

Dear reviewer:

Thank you very much for your professional comments on our articles. As you are concerned, there are several problems to be solved. According to your suggestions, we have made many changes to the previous manuscript. We will provide you with two documents, one of which is marked in pink after modification according to your question. On the other hand, based on your comments, I would like to make the following general reply:

Question 1:Introduction should be improved.

Reply 1:Thank you for reminding me that my introduction could not better introduce the background of the article. Therefore, we found more relevant literature from authoritative journals, introduced the background of the article in different aspects and improved the English level.

Question 2:Page 1, line 33: Botrytis cinerea should be written in italics (and throughout the whole manuscript as well). Similarly, other phrases like in vitro should follow the same rule.

Reply 2:We modified the words of the article by referring to the highlighted words or phrases on the PDF file, especially Botrytis cinerea.

Question 3:Page 1, line 37: What is “Revised text:” stands for in the manuscript?

Reply 3:Before modification:The application of fungicide spraying is typically efficacious in suppressing the disease.[4]

After modification,Page 1, line 41:The most common and conservative way to control the devastating damage of Botrytis cinerea to agriculture is to use large quantities of chemical fungicides, such as thiabendazole, phenylpyrrole, benomyl, and flunidoxime[9]. However, the excessive utilization of fungicide can readily engender drug resistance in certain pathogenic microorganisms[10].

Cause:After introducing Botrytis cinerea, it is necessary to tell the readers what the previous treatment methods are and what the disadvantages are, but my expression is not clear. Therefore, we searched for better literature and learned their expression methods, hoping that this writing style can make everyone understand my idea more clearly.

Question 4,Page 1, line 41: What AZOX stands for? Before the use of acronyms authors should mention the proper name and put the acronym in parenthesis.

Reply 4:Thank you for your suggestion.Page 1, line 13,Azoxystrobin (AZOX).

Question 5:Page 1, line 4: It has been a number of lead molecules that derived from a number of basidiomycetes, such as Strobilurus tenacellus, that were then developed to azoxystrobin, as we use it today, and other important fungicides and oomycetides. Thus, authors should develop more in details on this, since azoxystrobin plays a pivotal role in their research.

Reply 5:Thank you for your advice. Pyrimidin plays a key role in the research, so we look for more authoritative references, from different angles for more detailed elaboration.

Question 6:Page 2, lines 47-48: “The limited water solubility of AZOX (6.7μg/mL). [8] However, has significantly impeded its application in plant protection.”

To the best of my knowledge azoxystrobin has been the pioneer strobilurin fungicide and has been registered in numerous crops against major plant pathogens worldwide for over 20 years. Thus, I cannot accept the aforementioned statement of its impended application, especially if it is not supported by any relevant references, at least.

Reply 6:”The limited water solubility of AZOX (6.7μg/mL).”To prove the accuracy of the data, we cited more articles with the same point of view.

Question 7:Page 2, line 75: What BC stands for?

Reply 7:Biomass charcoal (BC),Page 1, line 17.

Question 8:Page 5, line 193: “2.8. Study on the bactericidal activity of potted plants”.

Botrytis cinerea is not a bacterium, but an ascomycete, a true fungus. Thus, the term bactericidal is not proper and should be replaced by the term fungicidal.

Reply 8:Thank you for your suggestion. “Study on the fungicidal activity of potted plants” is the most correct way to write it. After correction, we have modified similar contents in the whole paper.

Question 9:Page 10, paragraph 3.8: Authors present results regarding the in vitro activity on the mycelial growth od B. cinerea. However, their analysis is only qualitative, missing numerical, quantitative data and statistical analysis.

Reply 9:In order to express the results of in vitro activity more clearly, we added more data and other text contents in the text "3.4. In vitro fungicidal activity."

Question 10:Page 11, paragraph 3.9:

(1) In Fig. 9b azoxystrobin performs better to AZOX-ZIF-8/BC in terms of disease index, biologically meaning that with azoxystrobin we estimated less disease level compared to AZOX-ZIF-8/BC. They are statistically not significantly different only at the concentration of 500 ppm. This outcome is not clearly mentioned in the text of 3.9.

Question 11:(2) It is not clear the reverse of Fig 9b when interpreting the same data to Fig 9c at the concentrations of 100, 200, and 300 ppm. Please provide sufficient biological explanation.

Reply 10 and 11:I am sorry that I reversed the data of AZOX and AZOX-ZIF-8 /BC when making Figure 9b, so that you could not understand the content of the picture and text. For this, I remade the image and modified it with the text. Figures 9b and c are analyzed in more detail in "3.5. Study on the fungicidal activity of potted plants."

Question 12:(3) Authors in the text of 3.9 as well as in Fig. 9 present data without mentioning whether these data refer to preventative or therapeutic application of the active ingredient. In Paragraph 2.8 they write that they performed the experiment using both application regimes.

Reply 12:Not preventative or therapeutic application. The active ingredient plays a " control" effect in use. The word "control" can be found in the Chinese National Standard (NY/T 1156.10-2008).

Reviewer 6 Report

The work is for sure of interest because it presents an innovative approach to enhance pesticide efficiency while minimizing usage. Anyway I suggest of reviewing the work with some reconsideration as follows:

- In paragraph 2.2 "Synthesis of ZIF-8 and preparation of BC" the methods applied are developed by you or is already present in bibliography? If so please cite it.
- Line 127, maybe is better to specify the exact "varying ratios".
- In the results there is no correspondence with the numbering scheme of the paragraphs in materials and methods, therefore this makes more difficult understanding the results. For example in materials and methods the AZOX loading has numbering 2.3 while in results it has 3.2.  Therefore looking at the corrispondance of the paraghraps is required.

Author Response

Dear reviewer:

Thank you very much for your professional comments on our articles. As you are concerned, there are several problems to be solved. According to your suggestions, we have made many changes to the previous manuscript. We will provide you with two documents, one of which is marked in pink after modification according to your question. On the other hand, based on your comments, I would like to make the following general reply:

Question 1:- In paragraph 2.2 "Synthesis of ZIF-8 and preparation of BC" the methods applied are developed by you or is already present in bibliography? If so please cite it.

Reply 1:Thank you for your reminder, detailed explanation and references [31] and [32] on page 3, lines 108 to 123.

Question 2:- Line 127, maybe is better to specify the exact "varying ratios".

Reply 2:Thank you very much for your reminding. Here, we have modified “2. 2.3. Preparation of AZOX loaded ZIF-8/BC” and specified the exact "change ratio".

Question 3:- In the results there is no correspondence with the numbering scheme of the paragraphs in materials and methods, therefore this makes more difficult understanding the results. For example in materials and methods the AZOX loading has numbering 2.3 while in results it has 3.2.  Therefore looking at the corrispondance of the paraghraps is required.

Reply 3:Thank you for your comments. The confusion of numbering in the original article made it difficult for readers to understand. In order to solve this problem, we refer to article “Emulsion-based synchronous pesticide encapsulation and surface modification of mesoporous silica nanoparticles with carboxymethyl chitosan for controlled azoxystrobin release” from authoritative journal “Chemical Engineering Journal” and re-number it.

Round 2

Reviewer 2 Report

the authors addressed almost comments that were raised during the first revision.